

# Performance evaluation of the inverse real-valued fast Fourier transform on field programmable gate array platforms using open computing language

Li Liu*, Sida Yang*, Haoyu Tan, Fengzhan Zhou, Jiantao Yin, Zishen Cao, Tianhao Wang, Zhuo Qian and Guoyou Gan

Faculty of Material Science and Engineering, Kunming University of Science and Technology, Kunming, Yunnan, China
* These authors contributed equally to this work.

## ABSTRACT

The real-valued fast Fourier transform (RFFT) is well-suited for high-speed, low-power FFT processors, as it requires approximately half the arithmetic operations compared to the traditional complex-valued FFT (CFFT). While RFFT can be computed using CFFT hardware, a dedicated RFFT implementation offers advantages such as lower hardware complexity, reduced power consumption, and higher throughput. However, unlike CFFT, the irregular signal flow graph of RFFT presents challenges in designing efficient pipelined architectures. In our previous work, we have proposed a high-level programming approach using Open Computing Language (OpenCL) to implement the forward RFFT architectures on Field-Programmable Gate Arrays (FPGAs). In this article, we propose a high-level programming approach to implement the inverse RFFT architectures on FPGAs. By identifying regular computational patterns in the inverse RFFT flow graph, our method efficiently expresses the algorithm using a for loop, which is later fully unrolled using high-level synthesis tools to automatically generate a pipelined architecture. Experiments show that for a 4,096-point inverse RFFT, the proposed method achieves a 2.36x speedup and 2.92x better energy efficiency over CUDA FFT (CUFFT) on Graphics Processing Units (GPUs), and a 24.91x speedup and 18.98x better energy efficiency over Fastest Fourier Transform in the West (FFTW) on Central Processing Units (CPUs) respectively. Compared to Intel's CFFT design on the same FPGA, the proposed one reduces 9% logic resources while achieving a 1.39x speedup. These results highlight the effectiveness of our approach in optimizing RFFT performance on FPGA platforms.

# INTRODUCTION

The fast Fourier transform (FFT) is one of the most important benchmarking applications on many high-performance computing platforms. The conventional FFT algorithm has assumed an input sequence consisting of complex-valued numbers (CFFT). However, the

Corresponding authors
Zhuo Qian, 20210067@kust.edu.cn
Guoyou Gan, ganguoyou@kmust.edu.cn

input sequence consists of real-valued numbers (RFFT) in a large number of applications. Recently there has been an increasing interest in RFFT since most signals in the physical world are real-valued. For example, in medical applications, the RFFT algorithm is used to estimate the power spectral density of electroencephalography and electrocardiography (*Cheng et al., 2008*). In telecommunications, the RFFT is a key element in signal modulation, demodulation, and transmission (*Ibraheem & Khamiss, 2008*).

Even though the data points are real, CFFT algorithms can still be applied. One simple approach is to use real data for the real components and zeros for the imaginary components, and then the CFFT algorithms can be applied directly. However, this method is not efficient. When the inputs are real, the complex outputs of FFT algorithms exhibit the property of conjugate symmetry, and by exploiting this special property, approximately half of the arithmetic operations are redundant and could be removed (*Garrido, Parhi & Grajal, 2009*). The reduced arithmetic complexity can further lead to efficient hardware structures with less area and power consumption while offering high signal processing capabilities. Such specialized RFFT processors are of great importance in the design of wearable or implantable medical devices.

A primary challenge in designing efficient pipelined architectures for RFFT lies in the irregular geometries of the signal flow graph. A novel approach to designing a regular signal flow graph for RFFT was proposed for the first time in *Garrido, Parhi & Grajal (2009)*. This innovative technique systematically eliminates redundant operations within each stage of the signal flow graph and strategically relocates twiddle factors between stages to achieve regular geometries. Recent advancements, exemplified by *Ayinala & Parhi (2013)* and *Salehi, Amirfattahi & Parhi (2013)*, in the development of efficient pipelined architectures for RFFT draw inspiration from this seminal approach.

Nevertheless, prior efforts have predominantly focused on implementing the RFFT pipeline in a stage-by-stage fashion at register-transfer-level (RTL) using hardware description language (HDL). The development of an RFFT pipeline with HDL presents various challenges, including the necessity to design and verify extensive and intricate RTL code, leading to limited scalability. In fact, in *Garrido, Parhi & Grajal (2009)*, *Ayinala & Parhi (2013)*, and *Salehi, Amirfattahi & Parhi (2013)*, the reported maximum size of the design is limited to a 128-point RFFT. In addition, the results are theoretical and are based on a 16-bit word length. Furthermore, it is not trivial to incorporate FPGAs with HDL-based designs in frameworks that have other types of computing devices such as CPUs, GPUs or even DSPs, limiting the impact of using FPGAs in high-performance computing environments.

Recently emerged high-level synthesis (HLS) languages, such as C/C++ and OpenCL (*Khronos Group, 2016*), have the potential to overcome many of the limitations associated with traditional HDL-based development. Compared with hardware description languages, high-level design offers several advantages. First, it allows designers to concentrate on algorithm specification rather than the low-level details of hardware implementation, such as datapath construction, control circuitry, and timing closure. These tasks are instead handled automatically by the HLS compiler. Moreover, high-level designs can be more easily migrated to newer FPGA platforms with larger capacities and

higher performance, as the compiler can re-target the same high-level description to generate optimized hardware for the new device.

Building on these advantages, a number of studies have explored the application of HLS in FFT design. For instance, in *Wang et al. (2025)*, the authors developed a general FFT HLS generator that provides multiple functionalities and customizable parallelism settings to address diverse user requirements. In *Lee et al. (2024)*, HLS tools are employed to explore design alternatives and determine the optimal combination of architectural parameters, such as radix, memory ports, and loop/block pipelining. In *Xu et al. (2017)*, by explicitly specifying the target datapath in HLS code, the latency of a radix-2 FFT implementation was effectively reduced. Similarly, in *Almorin et al. (2022)*, the authors proposed a generic FFT behavioral model that incorporates different parallelization strategies. Despite these advances, relatively few studies have focused on leveraging HLS tools for implementing real-valued FFTs, leaving this area underexplored.

In this article, we use OpenCL to design efficient pipelined architectures that can be easily extended to process any RFFT size which is a power of 2 (depending on the available hardware resources on FPGAs). First, we obtain a signal flow graph consisting of only real datapaths with a relatively regular twiddle factor distribution. Second, we identify the fixed computation pattern in it and design corresponding butterfly structures. Furthermore, by building mathematical relationships among the stage number of the flow graph, the butterfly number in each stage and the memory addresses of twiddle factors, we propose novel twiddle factor access schemes for the RFFT. At last, we use a for loop to implement the entire RFFT algorithm and with the help of Intel's high-level synthesis tools, the loop can be fully unrolled to automatically construct pipelined architectures on FPGAs. This article intends to provide a design methodology that maintains the simplicity of high-level software programming while offering the speed of a dedicated hardware accelerator for RFFT calculation.

In the next section we develop the algorithm that allows the design of regular hardware architectures for the RFFT. The proposed design is evaluated and compared to some well-known benchmarks in 'Evaluation' and conclusions are drawn in 'Conclusion'.

## PROPOSED METHODOLOGY

In our previous work (*Qian & Gan, 2024*), we illustrated the calculation of the real-valued FFT (RFFT) using a high-level synthesis approach. In this article, we extend the approach by employing the real-valued inverse FFT (RIFFT) to further demonstrate the idea. The N-point inverse discrete Fourier transform is defined by

$$x(k) = \frac{1}{N} \sum_{n=0}^{N-1} X(n) W_N^{nk} \qquad (1)$$

where $k = 0, 1, \ldots, N-1$ and $W_N^{nk} = e^{j2\pi nk/N}$.

The basic idea behind the RIFFT is that, if the input sequence $X(n)$ exhibits the property of conjugate symmetry, *i.e.*,

$$X(N-n) = X^*(n). \qquad (2)$$

then the output $x(k)$ is real. By exploiting this property, only $(N/2) + 1$ inputs are required to compute the full output spectrum. The reader is referred to *Garrido, Parhi & Grajal (2009)* for details on state-of-the-art algorithms and pipelined architectures for real-valued FFT. Building on the idea of eliminating redundant operations, we propose a high-level programming methodology for developing pipelined architectures. The proposed methodology consists of three steps, which are explained below.

## Regular computational pattern

When the inputs are real, the method proposed in *Garrido, Parhi & Grajal (2009)* first eliminates redundant operations in the flow graph of the inverse CFFT (highlighted in the blue regions of Fig. 1). After relocating the twiddle factors between stages, a 32-point flow graph of the RIFFT obtained using this method is shown in Fig. 2. This flow graph consists entirely of real datapaths, and the input samples are annotated with a letter (r or i) to indicate whether the value corresponds to the real or imaginary part of the output.

Based on Fig. 2, we first remove the $W^0$ blocks (marked by red boxes) and replace them with the "X" symbol to achieve a more regular distribution of twiddle factors. These $W^0$ blocks, highlighted in the red boxes of Fig. 1, are not considered twiddle factors in the conventional CFFT signal flow graph; instead, they merely indicate that input data should be stored at different output locations. This modification results in the regular butterfly architectures between stage 1 and $log_2 N - 3$ shown in Fig. 3, where all datapaths are real. Second, we introduce a butterfly counter in each stage. The modified flow graph is presented in Fig. 3, with different colors used to highlight the butterfly structures involved.

In this article, to enable throughput comparisons with off-the-shelf FFT designs (*Intel, 2016c*), we employ butterfly units capable of processing eight real data points at a time. In Fig. 4, between stage 1 and $log_2 N - 2$, we observe that processing eight data points simultaneously requires two types of butterfly structures. The first type, shown in Fig. 4A, is used when the real transforms end and the complex sub-transforms begin. The block marked "X" indicates that no multiplication is required, and the data points are instead stored directly at different output locations. The second type, shown in Fig. 4B, is used within the complex sub-transform. It is worth noting that the regular computation pattern is not unique. For example, butterfly units that process four real data points at a time could also be used to construct such patterns.

We develop a scheduling algorithm for these two types of butterfly units. The algorithm determines which type of butterfly unit should be used in each stage. The equation for calculating the butterfly unit type is given as follows:

$$B/(1 \ll P) \tag{3}$$

where $N$ denotes the FFT size, $B$ denotes the butterfly number and $P$ denotes the stage number. If the above equation evaluates to 0, a type I unit is used; otherwise, a type II unit is selected. For example, in Fig. 3, all butterflies in stage 2 yield an equation value of 0; therefore, only type I units are used in this stage. In stage 1, butterflies 0 and 1 also yield 0, and thus use type I units, while butterflies 2 and 3 yield 1, and therefore use type II units.

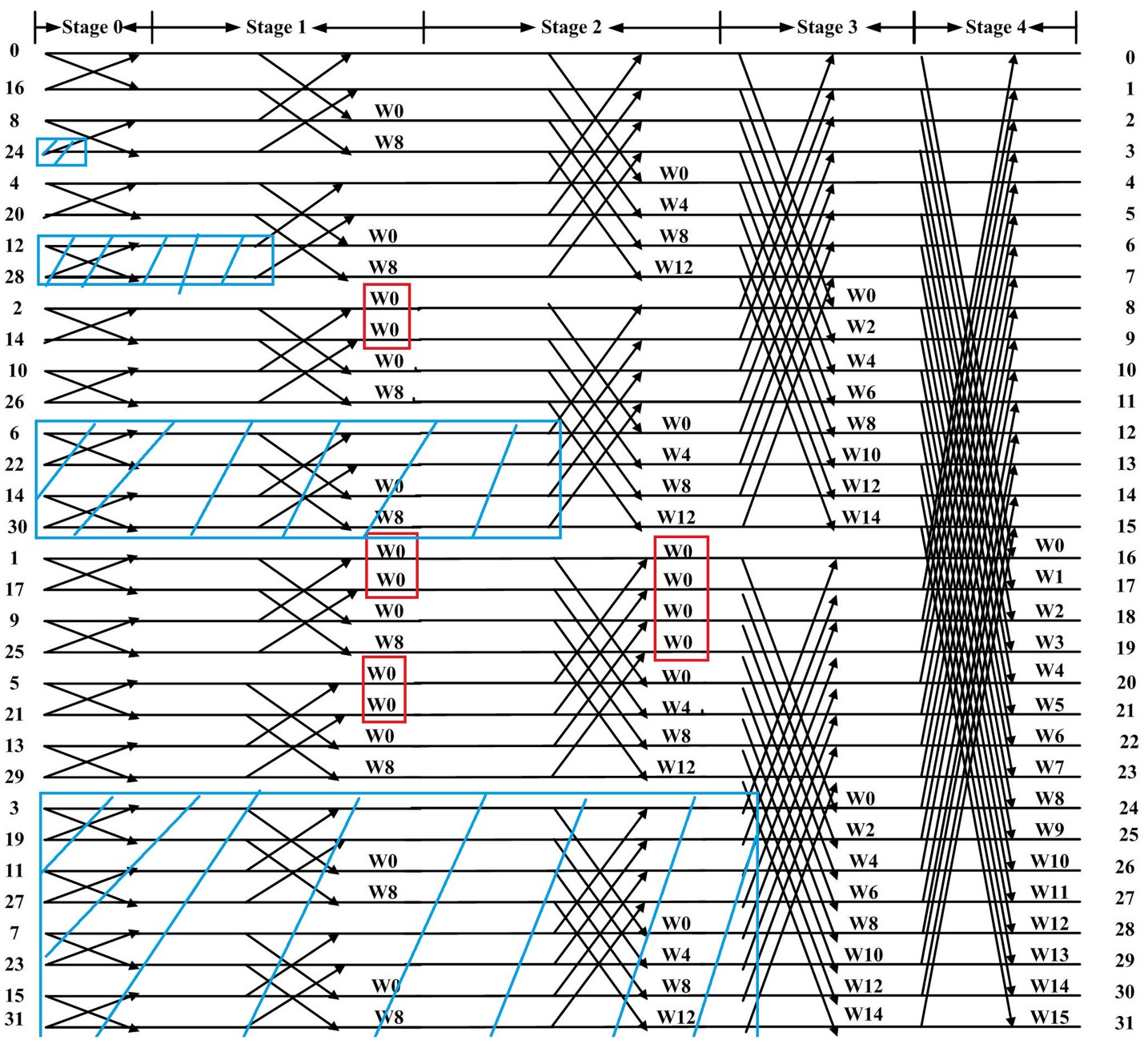

**Figure 1  Flow graph of 32-point DIT CFFT with blue regions indicating redundant operations.**

The overall structure of the proposed OpenCL kernel is shown in Fig. 5. Stage 0 and the last two stages have fixed computational patterns and can be hard-coded outside of the loop. The computations between stage 1 and $log_2N - 3$ are carried out using the two types of butterfly units shown in Fig. 4. The for-loop consists of three parts: (a) the butterfly scheduling algorithm described above. (b) the FFT data access scheme. (c) the twiddle factor access scheme. If these three components are properly designed, the loop can be fully unrolled (#pragma unroll) by high-level synthesis tools to construct pipelined
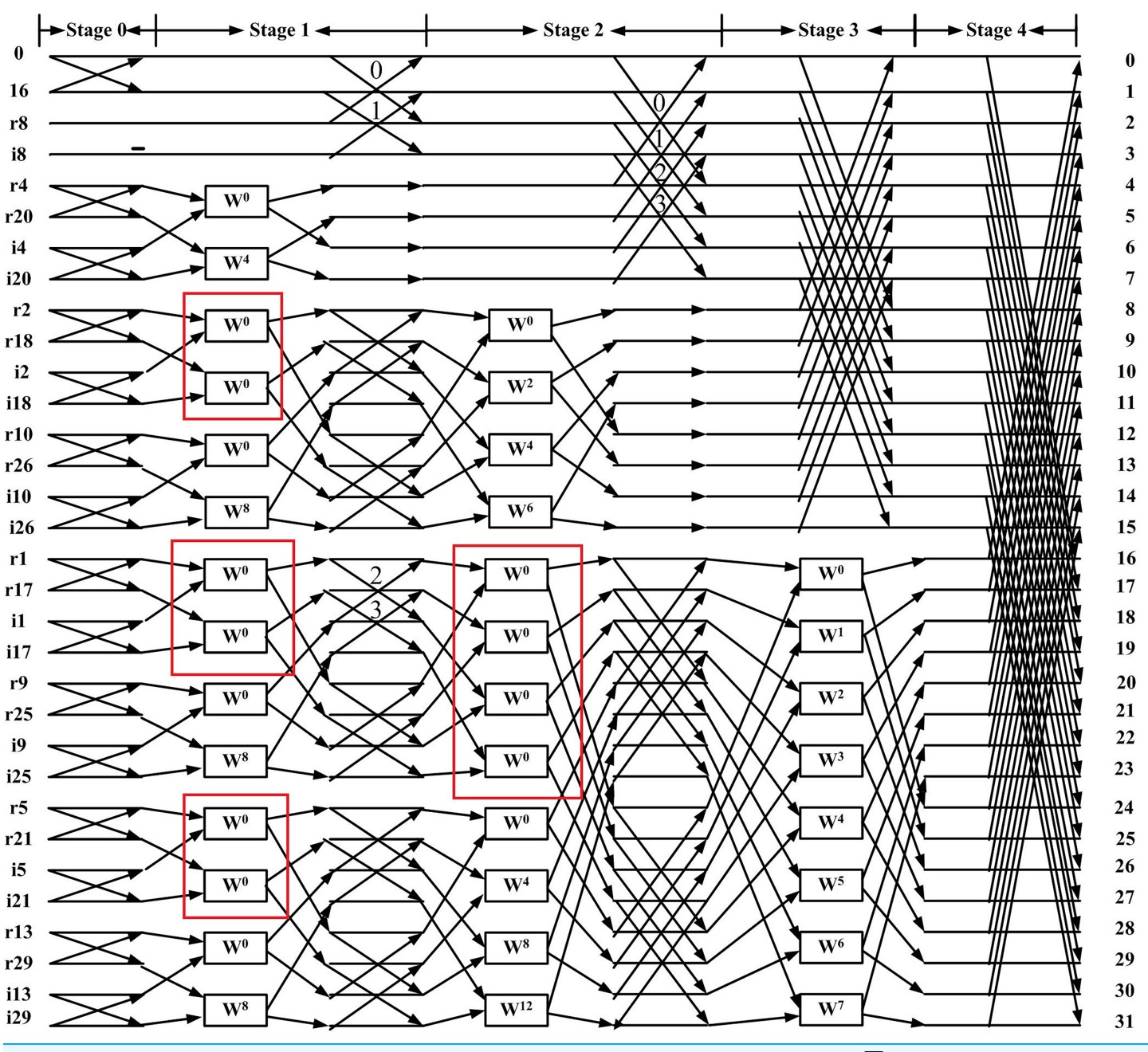

**Figure 2 Flow graph of 32-point DIT RIFFT (*Garrido, Parhi & Grajal, 2009*).**

architectures. In this case, the initiation interval (*Intel, 2016a*) equals one, indicating that the pipeline has no data or memory dependencies (*Intel, 2016b*) and is capable of computing one sample per clock cycle.

In the flow graph, we use a decimation-in-time (DIT) radix-8 RFFT to illustrate the idea, but similar approaches can also be applied to the decimation-in-frequency RFFT algorithm. Moreover, once the fixed computational pattern is extracted, other radices of

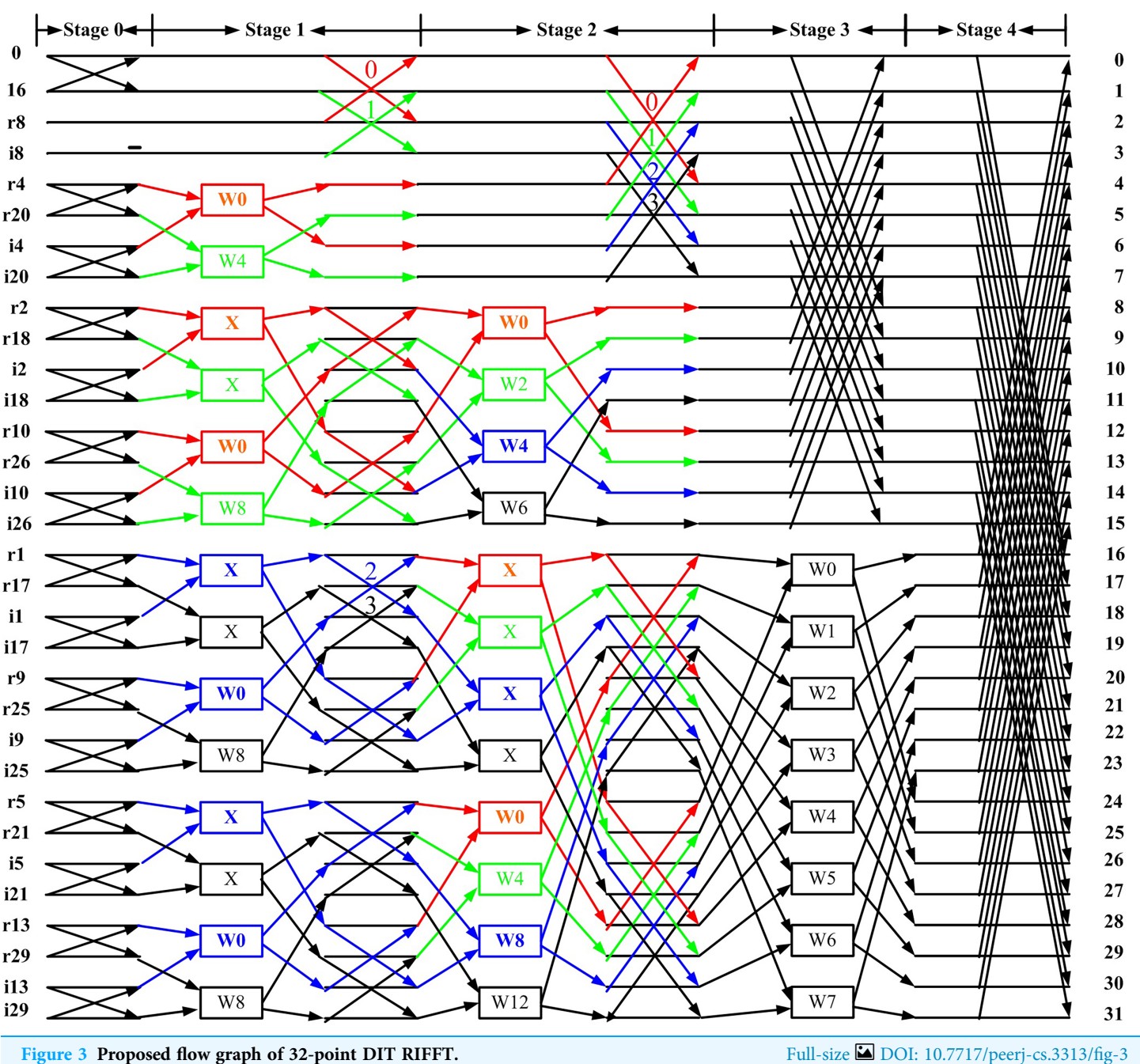

**Figure 3 Proposed flow graph of 32-point DIT RIFFT.**

butterflies, such as radix-2, can also be employed for computation, with corresponding data access schemes and twiddle factor access schemes developed accordingly.

## FFT data access scheme

We directly adopt the data access scheme described in *Intel (2016c)*. A sliding window and data reordering algorithm are designed to provide the butterfly unit with the correct FFT data in each clock cycle. The sliding window is essentially a buffer of length

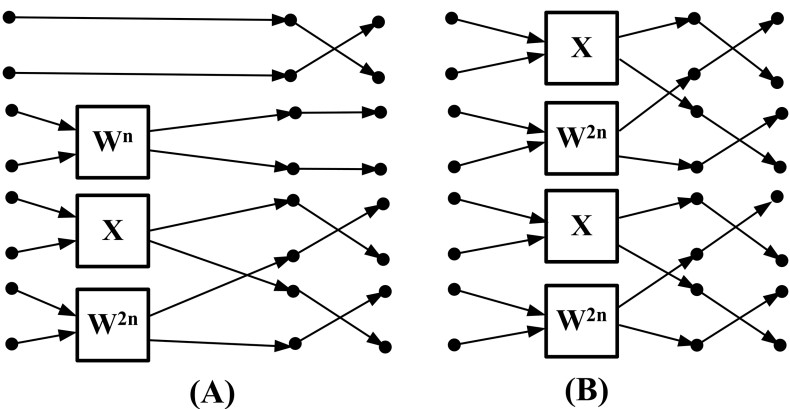

**Figure 4** Proposed butterfly structures: (A) type I unit (B) type II unit.

---
**OpenCL Kernel Structure**

---

1: Stage 0 of feed-forward FFT

2: #pragma unroll

3: for stage $\in [1, logN - 3]$ do

4:     Fetch FFT data

5:     Calculate butterfly type using equation (3)

6:     Calculate twiddle addresses using equation (4) and (5)

7:     perform butterfly operation

8: end for

9: Stage $logN - 2$ of feed-forward FFT

10: Stage $logN - 1$ of feed-forward FFT

---

**Figure 5** Pseudocode of the proposed kernel.

$N + 8 * (log_2 N - 2)$ and the data stored in this buffer are shifted by one element at each iteration. The reader is referred to *Intel (2016c)* for further details.

**Twiddle factor access scheme**

Twiddle factors are precomputed and stored in a ROM bank with consecutive addresses, as shown in Table 1. For example, $W^N$ is stored at the address $N$.

Based on the proposed scheduling algorithm and the structure of the butterfly units, we establish mathematical relationships among the stage number of the signal flow graph, the butterfly index in each stage, and the memory addresses of the twiddle factors. First, for the type I butterfly unit, the address of the upper twiddle factor is given by:

$$B * (1 \ll (log_2 N - P - 2)) \tag{4}$$

where $B$ denotes the butterfly number and $P$ denotes the stage number. For example, in Fig. 3 for butterfly 3 in stage 2, according to Eq. (3) it is a type I butterfly unit and the above equation evaluates to 6. This means the upper twiddle factor in the type I unit is $W^6$ and the lower twiddle factor is $W^{12}$. Similarly, for butterfly 1 in stage 1, it is also a type I unit

**Table 1 ROM configuration for a N-point RFFT.**

| Address | Twiddle factors |
|---|---|
| 0 | $W^0$ |
| 1 | $W^0$ |
| ... | ... |
| N − 1 | $W^{N-1}$ |

and the above equation evaluates to 4, indicating that the upper twiddle factor is $W^4$ and the lower twiddle factor is $W^8$. For the type II butterfly unit, the address of the upper twiddle factor is given by

$$(B\%(1 \ll P)) * (1 \ll (log_2N - P - 1)). \tag{5}$$

For example, for butterfly 3 in stage 1, it is a type II butterfly unit and the above equation evaluates to 8. This means both the upper and lower twiddle factor in the type II unit is $W^8$.

# EVALUATION

To show the performance and energy efficiency advantages of the proposed RFFT framework, we conducted experiments over various benchmarks and hardware platforms. To the best of our knowledge, our work is the first real FPGA board implementation of RIFFT.

## Experiment setup

The importance of FFT in DSP applications has led to a number of well-known benchmarks.

*CPU benchmark:* FFTW (*Massachusetts Institute of Technology, 2016*) is a widely used free-software library that computes the discrete Fourier transform and its various special cases on scalar processors like CPUs. Its performance is competitive with vendor-optimized programs, but unlike these programs, FFTW is not tuned to a fixed machine. FFTW provides separate and specialized routines for calculating both CFFT and RFFT.

*GPU benchmark:* The CUFFT (*Nvidia Developer, 2016*) is the NVIDIA CUDA FFT product. It provides a simple interface for computing FFTs on an NVIDIA GPU, which allows users to quickly leverage the floating-point power and parallelism of the GPU. Contrary to FFTW, CUFFT is designed for vector processors like GPUs and hence it is much faster than FFTW. CUFFT offers several computation modes: C2C (complex to complex), C2R (complex to real) and R2C (real to complex). C2C is the regular complex FFT, C2R is the RFFT algorithm and R2C is the inverse of the RFFT, which could be implemented using a similar strategy presented in this article.

*FPGA benchmark:* Intel has an OpenCL implementation of one-dimensional CFFT (*Intel, 2016c*). The example processes 4,096 complex single-precision floating-point values. The input data is ordered and the output data is in bit-reversed order. The example contains a single radix-4 butterfly unit capable of processing 4 complex data points (eight real data points) per clock cycle. In order to compare with this benchmark, in our proposed

design we have also chosen to use butterfly units capable of calculating eight real data points per clock cycle.

The OpenCL kernel which contains the proposed algorithm is pre-compiled using Intel's OpenCL tool kits. The OpenCL host program is running on the CPU and the kernel is running on the FPGA. When compiling the kernel, three optimization techniques are used: (1) Loop unrolling (*Intel, 2016d*): loop unrolling involves replicating a loop body multiple times and decreases the number of iterations at the expense of increased hardware resource consumption. We use the "#pragma unroll" directive to unroll the loop in Fig. 4 and the compiler automatically builds the pipeline. The data or memory dependency (*Intel, 2016b*) is removed and the initialization interval is equal to one (*Intel, 2016a*). (2) Optimize floating point operations: with the "–fpc" (*Intel, 2016e*) compilation flag turned on, the Intel OpenCL compiler tries to remove floating point rounding operations and conversions whenever possible. (3) Non-interleaved memory (*Intel, 2016f*): in our case, we want to partition the banks manually as two non-interleaved (and contiguous) memory regions, using the "-no-interleaving" compilation flag, to achieve better load balancing. Specifically, in this work the kernel will read input data from one off-chip RAM bank and store the results in the other RAM bank. During computation all the intermediate data points are stored on the FPGA chip and the kernel does not have to interact with the off-chip RAM multiple times. By configuring FPGA memory in this way, the Von Neumann bottleneck is minimized.

## Results

In this section, we compare the proposed RFFT with FFTW, CUFFT and Intel's CFFT.

Performance tests of our proposed RFFT algorithm are carried out on a Nallatech PCIe-385N A7 FPGA board and it is compared to Intel's CFFT on the same board. We also compare our implementation to FFTW and CUFFT tested on a laptop with an Intel Core i7-4510U CPU running at 2 GHz and a Nvidia GeForce GT 720M GPU running at 1.4 GHz. The kernel execution time of various FFT sizes and platforms is reported in Table 2. There are generally two ways to benchmark FFT. One approach is to use *points/sec*, defined by *N/(time for one FFT in seconds)*. The other is to use *flops/sec*, defined by *$5N\log_2 N$/(time for one FFT in seconds)*. Our results match closely to Intel's reported results in *Intel (2016c)*. For a 4,096-point CFFT, Intel has reported a throughput of 81 GFLOPS on the BittWare S5-PCIe-HQ D5/D8 board while our result is 83 GFLOPS on a different board. For a 4,096 input size, in terms of *points/sec*, the proposed RFFT achieves approximately a 1.39x speedup compared with Intel's CFFT, a 2.36x speedup compared with the RIFFT routine of CUFFT and a 24.91x speedup compared with the RIFFT routine of FFTW.

Xilinx has a document regarding their implementations of CFFT using HDL (*Xilinx, 2011*). The implementation details outlined in *Xilinx (2011)* and *Salehi, Amirfattahi & Parhi (2013)* differ from ours in several aspects. First, these two documents utilize Virtex family FPGAs while we utilize Stratix-V FPGAs. Second, these documents utilize a 16-bit or 20-bit word length to represent data while we employ a single-precision floating-point data format. Consequently, our calculated results can be effortlessly validated against FFTW or CUFFT. Despite these major differences, a rough comparison is still feasible. In

**Table 2  Kernel execution time (ms) of 2,000 runs of various FFT benchmarks.**

| FFT size | 32 | 64 | 128 | 256 | 512 | 1,024 | 2,048 | 4,096 |
|---|---|---|---|---|---|---|---|---|
| Proposed RIFFT on FPGA | 0.203 | 0.220 | 0.275 | 0.371 | 0.684 | 1.342 | 2.131 | 4.251 |
| Intel's CFFT on FPGA | 0.272 | 0.278 | 0.316 | 0.551 | 0.939 | 1.627 | 3.403 | 5.904 |
| Nvidia's RIFFT on GPU | 0.168 | 0.231 | 0.375 | 0.689 | 1.321 | 2.565 | 5.086 | 10.026 |
| Nvidia's CFFT on GPU | 0.1207 | 0.1845 | 0.3436 | 0.6644 | 1.2927 | 2.5620 | 5.0262 | 9.9345 |
| FFTW's RIFFT on CPU | 0.508 | 0.869 | 2.763 | 6.632 | 15.931 | 30.100 | 55.136 | 105.768 |
| FFTW's CFFT on CPU | 0.979 | 2.734 | 5.799 | 15.725 | 32.022 | 55.696 | 89.132 | 151.771 |

**Table 3  Resource utilization ratio of proposed RIFFT and Intel's CFFT on nallatech pcie-385n a7 device (RIFFT/CFFT).**

| FFT size | 32 | 64 | 128 | 256 | 512 | 1,024 | 2,048 | 4,096 |
|---|---|---|---|---|---|---|---|---|
| Logic utilization | 34%/39% | 36%/41% | 38%/46% | 40%/46% | 42%/50% | 44%/51% | 46%/55% | 48%/57% |
| Dedicated logic registers | 15%/17% | 16%/18% | 17%/20% | 17%/20% | 18%/22% | 19%/23% | 20%/24% | 21%/25% |
| Memory block | 31%/30% | 31%/30% | 31%/32% | 32%/32% | 32%/32% | 33%/33% | 33%/34% | 34%/34% |
| DSP block | 11%/19% | 16%/19% | 20%/28% | 25%/28% | 30%/38% | 34%/38% | 39%/47% | 44%/47% |

**Table 4  Power consumption of different devices.**

| Devices | Average power in watts |
|---|---|
| CPU | 16 |
| GPU | 26 |
| FPGA | 21 |

*Xilinx (2011)*, the maximum frequency is around 400 MHz for a 1,024-point CFFT, resulting in a theoretical throughput of 400 *MPoints/sec*. In contrast, according to Table 2 in Intel's original implementation, the throughput for a 1,024-point CFFT is 1,259 *MPoints/sec*, approximately three times faster than XFFT. In *Salehi, Amirfattahi & Parhi (2013)*, the article reports a throughput of 922 *MPoints/sec* for a 64-point RFFT using the radix-8 algorithm and two levels of parallelism. In our experiments, we employ a single instance of the kernel without replicating the pipeline (*Intel, 2016g*).

Details regarding the hardware resources available on the target FPGA board, including the count of DSP blocks and memory blocks, are provided in reference (*Nallatech, 2016*). The resource utilization ratio of the proposed RIFFT and Intel's CFFT on the same FPGA is reported in Table 3, demonstrating that the RIFFT algorithm consumes fewer hardware resources for each input size. Table 4 shows the average power consumption in different configurations. The proposed method achieves a 2.92× better energy efficiency over CUFFT on the GPU, and a 18.98× better energy efficiency over FFTW on the CPU respectively. Within the specified input range, FPGA exhibits clear advantages over GPU and CPU in terms of both throughput and power efficiency.

But it is also important to realize that while OpenCL or C/C++ enables device portability, achieving true performance portability requires careful tuning and adaptation to the underlying hardware architecture. OpenCL is designed as a cross-platform programming model, allowing the same kernel code to be compiled and executed on CPUs, GPUs, FPGAs, DSPs, and other accelerators from vendors such as Intel, AMD, NVIDIA, and Xilinx. This is achieved by abstracting the underlying hardware through a standard API and runtime. However, while the same code can run across devices, it rarely achieves comparable performance everywhere, since efficiency depends strongly on hardware-specific characteristics. For instance, the proposed kernel structure leverages pipelined parallelism, which is well suited for FPGAs. In contrast, on GPUs the kernel should be restructured to exploit single instruction, multiple data (SIMD) parallelism in order to fully utilize the architecture.

## CONCLUSION

This article describes a high-level design approach for the OpenCL implementation of the RIFFT algorithm on the CPU-FPGA platform. We identified the necessary radix-8 butterfly structures to compute RIFFT with any input size which is a power of 2. Corresponding twiddle factor addressing schemes are also developed. This work shows that the reduction of operations in RIFFT is not only theoretical but can also be applied to the design of efficient hardware architectures which require significantly fewer resources than CFFT implementations. In addition, one advantage of using OpenCL is that depending on the end application and the available FPGA resources, more instances of this kernel can be instantiated for higher performance.

## FUTURE WORK

As mentioned in the previous section, additional kernel instances can be instantiated to achieve higher performance. This opens up several potential research directions, such as investigating alternative radices, developing dynamic scheduling algorithms, and adapting the design to diverse FPGA architectures. Another promising direction is to extend the proposed approach to its 2D and 3D counterparts, as 2D and 3D RFFT/RIFFT operations are widely employed in Fourier convolutional neural networks.

### Funding

This research was funded by National Natural Science Foundation of China (Grant No. 62462039); the Major Science and Technology Project of Yunnan Province (Grant No. 202402AB080005); the Science and Technology Projects of Yunnan Precious Metals Laboratory (Grant No. YPML-2023050206); the Major Science and Technology project of Yunnan Precious Metals Laboratory (Grant No. YPML-20240502102). The funders had no role in study design, data collection and analysis, decision to publish, or preparation of the manuscript.

## Grant Disclosures

The following grant information was disclosed by the authors:

National Natural Science Foundation of China: 62462039.

Major Science and Technology Project of Yunnan Province: 202402AB080005.

Science and Technology Projects of Yunnan Precious Metals Laboratory: YPML-2023050206.

Major Science and Technology Project of Yunnan Precious Metals Laboratory: YPML-20240502102.

## Competing Interests

The authors declare that they have no competing interests.

## Author Contributions

- Li Liu conceived and designed the experiments, authored or reviewed drafts of the article, and approved the final draft.
- Sida Yang conceived and designed the experiments, performed the experiments, authored or reviewed drafts of the article, and approved the final draft.
- Haoyu Tan analyzed the data, prepared figures and/or tables, and approved the final draft.
- Fengzhan Zhou performed the computation work, prepared figures and/or tables, and approved the final draft.
- Jiantao Yin performed the computation work, prepared figures and/or tables, and approved the final draft.
- Zishen Cao performed the computation work, prepared figures and/or tables, and approved the final draft.
- Tianhao Wang performed the computation work, prepared figures and/or tables, and approved the final draft.
- Zhuo Qian conceived and designed the experiments, prepared figures and/or tables, authored or reviewed drafts of the article, and approved the final draft.
- Guoyou Gan conceived and designed the experiments, prepared figures and/or tables, authored or reviewed drafts of the article, and approved the final draft.

## Data Availability

The code and data are available at GitHub and Zenodo:

- https://github.com/memcpu-ZQ/FPGAFFT.
- memcpu-ZQ. (2025). memcpu-ZQ/FPGAFFT: RFFT on FPGA (v1.0.0). Zenodo. https://doi.org/10.5281/zenodo.17288743.

The data is available in the Supplemental File.

## Supplemental Information

Supplemental information for this article can be found online at http://dx.doi.org/10.7717/peerj-cs.3313#supplemental-information.

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
