# Peer review of "Performance evaluation of the inverse real-valued fast Fourier transform on field programmable gate array platforms using open computing language"

_PeerJ Computer Science, doi:10.7717/peerj-cs.3313_

## Round 0.1 · original submission · Major Revisions

· Academic Editor

Major Revisions

The three reviewers have a positive opinion of the submission. Addressing some of their comments may require a rather substantial amount of work, which is why I believe that a major revision of the manuscript may be needed.

**Language Note:** The review process has identified that the English language must be improved. PeerJ can provide language editing services - please contact us at [email protected] for pricing (be sure to provide your manuscript number and title). Alternatively, you should make your own arrangements to improve the language quality and provide details in your response letter. – PeerJ Staff

Reviewer 1 ·

Basic reporting

Although the manuscript is written in generally clear and professional English, occasional awkward phrasing and grammatical inconsistencies (e.g., verb tense shifts and article usage) could benefit from proofreading or light language editing for full clarity.

Although the introduction provides a solid context and references key prior work, the background could be strengthened by incorporating more recent or diverse sources particularly involving modern FPGA platforms or alternative high-level synthesis methods.

The paper is well-organized, with clear figures and, however, it lacks same approach in tables. It also lacks explicit mention or sharing of raw data (e.g., source code, FPGA configuration files), which limits full reproducibility of the experimental results.

Although the manuscript is largely self-contained and addresses its stated goal of evaluating inverse RFFT on FPGA; some performance claims (e.g., energy efficiency gains) could be better substantiated with more detailed experimental methodology or measurement protocols.

The paper does not rely on formal theorems, it defines all key computational structures and design equations clearly. On the other hand, the lack of mathematical rigor or formal validation (e.g., numerical accuracy analysis) may limit its depth for some theoretical audiences.

Experimental design

The manuscript presents original research on high-level FPGA implementation of inverse RFFT using OpenCL, which aligns well with the journal’s focus on computational methods. The novelty claim would be stronger with clearer differentiation from existing OpenCL-based industrial or academic implementations.

The research question is well articulated and addresses a significant gap in scalable, high-level design of RIFFT architectures for FPGAs. On the other hand, the claim of being the first real-board implementation would benefit from broader benchmarking or review of unpublished industrial work.

Although the study uses multiple platforms and industry-standard benchmarks to evaluate performance and energy efficiency, demonstrating technical rigor, the methodology would be more robust with detailed explanations of power measurement protocols and inclusion of statistical variability in results.

Although the methodology is clearly described at the algorithmic level, with step-by-step design logic and architectural explanation, reproducibility is limited.

Validity of the findings

The paper does not emphasize novelty as a criterion, it provides a clear rationale for replicating and improving RIFFT implementations on FPGAs using high-level synthesis. On the other hand, the contribution would benefit from deeper comparison to similar prior replication or optimization efforts to better contextualize its value to the literature.

While the study presents extensive performance benchmarks across various platforms with consistent input sizes and configurations, supporting robustness, it lacks raw data sharing, confidence intervals, or detailed explanations for the tables.

The paper concludes with allignment of research question, and appropriately grounded in the reported performance and resource utilization results. However, some claims on energy efficiency and scalability would be supported by deeper experimental validation or cautionary framing.

Additional comments

While the method is theoretically scalable, experiments are only shown for up to 4096-point RIFFT. Further exploration on larger FFT sizes (e.g., 8192, 16384) would be valuable.

The CPU and GPU used (Intel i7-4510U, GT 720M) are relatively old/entry-level hardware.

For high-confidence performance comparisons, testing against modern processors and GPUs would strengthen the findings.

Reviewer 2 ·

Basic reporting

The paper presents a solid contribution to the field of high-performance computing for signal processing, specifically focusing on the inverse Real-Valued Fast Fourier Transform (RIFFT) on FPGAs using OpenCL. The methodology is clearly outlined, and the performance comparisons against CPU and GPU benchmarks, as well as an Intel CFFT FPGA design, are valuable. The emphasis on high-level synthesis (HLS) and energy efficiency is timely and relevant.
Although the references are somehow old, they are generally appropriate and reflect the state of the art relevant to the paper's scope.

Experimental design

However, there are issues that the authors should consider.
• While the paper states that "fixed computation pattern" is identified and corresponding butterfly structures are designed, the explanation of how these two specific butterfly units (Fig. 4a and 4b) are derived from the overall RIFFT flow graph (Fig. 3) could be more explicit. A more detailed derivation or illustrative steps showing the transformation from the irregular graph to the regular patterns would enhance understanding. A clearer explanation of why these specific stages behave differently or why other stages are handled outside the loop (stage 0 and last two stages) would be beneficial.
• The equations for twiddle factor address (4 and 5) are provided, but the underlying logic or derivation for these relationships could be elaborated further. A visual aid or step-by-step example for a small N might clarify how these equations ensure correct twiddle factor access for the proposed butterfly units across different stages.
• The paper mentions that the Xilinx CFFT implementation [16] and other RFFT designs [5] utilize different FPGA families and data word lengths (e.g., 16-bit or 20-bit fixed-point vs. single-precision floating-point). While acknowledging these differences, a more detailed discussion on the implications for direct comparison is warranted. Floating-point operations inherently consume more resources and typically have lower throughput than fixed-point for the same operation, so stating the performance relative to fixed-point designs requires careful contextualization. While the comparison to Intel's CFFT on the same FPGA is strong, the comparison to Xilinx and others could be more nuanced.
• While the flow graphs (Fig. 1, 2, 3) are crucial, they are quite dense. Improving the resolution or potentially breaking down complex sections of Fig. 3 (Proposed flow graph) into smaller, more digestible sub-figures could aid readability, especially when trying to follow the different colored butterfly structures.
• Although OpenCL provides a higher level of abstraction, providing pseudocode for the core OpenCL kernel or a more explicit algorithmic description would greatly aid reproducibility for other researchers attempting to replicate or build upon this work.
• A dedicated "Future Work" section would be beneficial. The conclusion mentions that more kernel instances can be instantiated, expanding on other potential research directions (e.g., exploring different radices, dynamic scheduling, or adapting to different FPGA architectures) would add value.

Validity of the findings

The results presented in the paper appear valid based on the methodology and comparisons provided.

Reviewer 3 ·

Basic reporting

1. The paper is written in clear and professional English, with technically accurate terminology. However, there are instances where the phrasing could be simplified for better readability (e.g., long sentences in the introduction). Minor grammatical and stylistic improvements are needed.
2. the literature review could benefit from a broader discussion of recent FPGA-based high-level synthesis approaches to FFT to situate the contribution more strongly.

Experimental design

1. Most implementation details are described clearly (FPGA board, compiler flags, optimization techniques). However, more information about the exact hardware setup (e.g., FPGA clock frequency, board configuration parameters) would improve reproducibility. Providing kernel source code or pseudocode as supplementary material would strengthen replicability.

Validity of the findings

Data & Results: Results are systematically presented, comparing performance (execution time, throughput), resource utilization, and energy efficiency against multiple benchmarks. The data appear robust and consistent.

Analysis: The performance improvements (e.g., 2.36× over GPU CUFFT, 24.91× over CPU FFTW) are clearly quantified. The conclusions are supported by the results.

Limitations: The study is limited to a single FPGA platform (Nallatech PCIe-385N A7). Generalization to other FPGA families (e.g., Xilinx) is not explored. Authors acknowledge differences with existing implementations but a clearer discussion of scalability across devices would improve validity.

Additional comments

Strengths:

Addresses a relevant gap (inverse RFFT FPGA implementation using OpenCL).

Provides a complete evaluation against CPU/GPU/FPGA baselines.

Strong methodological description (flow graphs, butterfly scheduling).

Results demonstrate significant improvements in performance and energy efficiency.

Weaknesses:

Literature review could include more recent FPGA high-level synthesis works.

Some technical details (clock speed, exact FPGA resource breakdown, source code availability) are missing, limiting full reproducibility.

Discussion of limitations and portability across platforms could be expanded.

Minor English language polishing required.

---

## Round 0.2 · accepted · Accept

· Academic Editor

Accept

As the reviewers are satisfied with this revision, I am glad to accept this manuscript for publication in PeerJ Computer Science.

Reviewer 1 ·

Basic reporting

The modified version of the paper conforms all the requirements in basic reporting part.
The language is clear and unambiguous. Literature and references are enough for the topic. The structure of the article is more professional. Paper is self-contained with relevent results to hypotheses. Results are formal.

Experimental design

The modified version of the paper makes clear the understanding of the experiments. The addition of the source code improves the reproducibility of the study.
The paper conforms requirements of originality of the journal. Research question is well defined, relevant and meaningful. Method i described with sufficient detail.

Validity of the findings

The validity of findings are double checked with the modified version of the paper.
Imapact and novelty are well demonstrated. All underlying data is provided and robust. Conclusions are well stated.

Reviewer 2 ·

Basic reporting

The authors have properly addressed the issues.

Experimental design

Although there are some limitation in conducting the experiments, however, the clarifications are sufficient

Validity of the findings

The results and valid and reliable